# DELTASCORE: Fine-Grained Story Evaluation with Perturbations

**Zhuohan Xie**   **Miao Li**   **Trevor Cohn**[*]   **Jey Han Lau**

School of Computing and Information Systems,
The University of Melbourne
{zhuohanx, miao4}@student.unimelb.edu.au, {t.cohn, laujh}@unimelb.edu.au

## Abstract

Numerous evaluation metrics have been developed for natural language generation tasks, but their effectiveness in evaluating stories is limited as they are not specifically tailored to assess intricate aspects of storytelling, such as fluency and interestingness. In this paper, we introduce DELTASCORE, a novel methodology that uses perturbation techniques for the evaluation of nuanced story aspects. We posit that the extent to which a story excels in a specific aspect (e.g., fluency) correlates with the magnitude of its susceptibility to particular perturbations (e.g., the introduction of typos). Given this, we measure the quality of an aspect by calculating the *likelihood difference* between pre- and post-perturbation states using pre-trained language models. We compare DELTASCORE with existing metrics on storytelling datasets from two domains in five fine-grained story aspects: fluency, coherence, relatedness, logicality, and interestingness. DELTASCORE demonstrates strong performance, revealing a surprising finding that one specific perturbation proves highly effective in capturing multiple aspects. Source code is available on our GitHub repository.[1]

## 1 Introduction

The emergence of large pre-trained language models (PLMs) (Zhao et al., 2023) has empowered story generation models to generate plausible narratives (Xie et al., 2021; Tan et al., 2021; Zhang et al., 2022b; Yang et al., 2022). The most advanced models have achieved the ability to produce stories which are not easily distinguishable from human-authored ones (Karpinska et al., 2021; Dou et al., 2022; Xie et al., 2023). However, the development of automated evaluation metrics in this domain has not progressed at the same pace (Guan et al., 2021b). Human evaluation, though

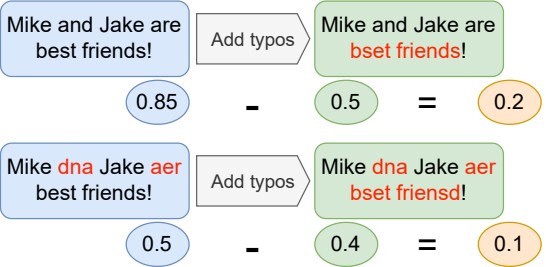

(a) Perturbation "Add typos" affects the highly fluent story (top) more than the less fluent one (bottom).

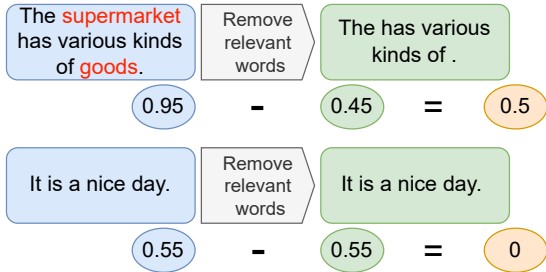

(b) Two stories are conditioned on the same title "I always go to the local supermarket". Perturbation "Remove relevant words" affects the highly related story (top) more while not affect the unrelated one (bottom).

Figure 1: Scenarios where higher quality stories (top) are affected more than lower quality ones (bottom) through aspect-specific perturbations (fluency: "Add typos"; relatedness: "Remove relevant words"). Generative likelihood for original/perturbed story is in blue/green circle, and the DELTASCORE value is in orange circle.

considered the gold standard, is hindered by its time-consuming, costly, and non-reproducible nature (Sai et al., 2023). Consequently, there is a demand for better automatic methods that can evaluate the quality of stories.

The prevailing evaluation metrics for story assessment have primarily been adapted from other natural language generation (NLG) tasks, such as BLEU (Papineni et al., 2002) for machine translation, or ROUGE (Lin, 2004) for summarization. Fortunately, recent progress has given rise to the emergence of new metrics explicitly tailored for

---

[*]Now at Google DeepMind
[1]https://github.com/ZhuohanX/DeltaScore

story evaluation, with a focus on quantifying story coherence (Guan and Huang, 2020; Ghazarian et al., 2021) or capturing human preferences (Chen et al., 2022). Other works have directly utilized the likelihood of a story under a PLM (Vaswani et al., 2017; Han et al., 2022) or its conditional likelihood based on human references or other contextual factors, such as story title (Thompson and Post, 2020; Yuan et al., 2021). Nonetheless, these approaches often yield a singular score that provides an estimate of the overall quality. However, Chhun et al. (2022) argue that the quality of a story is comprised of various fine-grained aspects, such as fluency and adherence to commonsense, suggesting that an overall quality score has limited utility for comprehensive story evaluation.

In this paper, we present DELTASCORE, a method that evaluates story quality by measuring the *likelihood difference* using a PLM between an original story and its perturbed version. The idea is that higher quality stories will exhibit more significant effects from the perturbation compared to lower quality ones. To provide fine-grained assessment of story quality, we experiment with perturbations that target specific aspects. Figure 1 presents two examples to demonstrate the intuition of our approach: 1) When we introduce random typos to modify the two stories shown in Figure 1a, we observe that the story with higher fluency is affected more by the perturbation; 2) When we modify the two stories in Figure 1b by removing relevant words, we observe that the perturbation affects the story that has a closer association with the title to a greater extent. Our empirical analysis demonstrates the superior performance of DELTASCORE compared to existing metrics in evaluating intricate story aspects. Furthermore, our investigation reveals an interesting discovery: one of our simplest perturbation methods, which simply shuffles all the words in the story, is very effective in capturing multiple aspects. This points to a possible interpretation that the pertubation may be functioning as a normalisation factor to modulate the effects of word frequency and text length when estimating sequence likelihood.

## 2  Related Work

### 2.1  Automatic Evaluation Metrics

Existing automatic evaluation metrics can be broadly categorized into three paradigms.

**Similarity metrics** mainly focus on measuring lexical overlap such as BLEU (Papineni et al., 2002), NIST (Doddington, 2002) and ROUGE (Lin, 2004) or semantic similarity with contextual representations including MoverScore (Zhao et al., 2019) and BERTScore (Zhang et al., 2020) between the machine-generated text and its human reference.

**Discriminative metrics** typically involve training a discriminator model to differentiate between high-quality and low-quality texts, including UNION (Guan and Huang, 2020), MANPLTS (Ghazarian et al., 2021), CTC (Deng et al., 2021), StoryER (Chen et al., 2022), and UNIEVAL (Zhong et al., 2022). Specifically, UNION constructs negative samples of original stories using heuristic rules and trains a discriminator to differentiate them. MANPLTS is an extension of UNION that constructs improved negative samples by manipulating storylines and generating alternate stories based on these manipulated storylines using a story generation model. StoryER builds a classifier to learn human preference by training it to differentiate highly-upvoted stories from lowly-upvoted ones on Reddit. CTC treats the evaluation task as an information alignment task. UNIEVAL frames the evaluation as a question answering task where different questions are asked to assess a particular aspect.

**Generative metrics** usually rely on generative likelihood to determine the quality of the text, including BARTScore (Yuan et al., 2021), T5Score (Qin et al., 2022) and GPTScore (Fu et al., 2023). Specifically, BARTScore evaluates generated text by calculating its conditional likelihood under BART. GPTScore calculates the likelihood of the story under a PLM with additional prefix to target a particular aspect. T5Score benefits from both worlds by employing both generative training with the standard negative log likelihood loss and discriminative training with contrastive loss where human judgments for generation quality are available.

### 2.2  Natural Text Perturbation

The use of perturbations is a conventional technique to generate negative samples for both discriminative (Guan and Huang, 2020) and generative (Zhong et al., 2022) tasks. Ribeiro et al. (2020) propose CheckList, a suite of perturbation techniques to evaluate the behavioral performance of

NLP models. Sai et al. (2021) further delve into applying perturbations to assess robustness of NLG evaluation metrics, while Karpinska et al. (2022) specifically focus on machine translation evaluation. He et al. (2022) also develop perturbation tests to identify blind spots of model-based evaluation metrics. Notably, all of these perturbations rely on heuristic rules. In contrast, recent adversarial attacks such as those proposed by Li et al. (2020); Morris et al. (2020) use language models to generate adversarial examples, which can also be considered a form of text perturbation. In our work, we explore perturbation for a different purpose: to evaluate fine-grained story qualities.

## 3   DELTASCORE

We now describe the idea of our approach. Given a story condition (e.g., a story title) $c = c_1, ..., c_n$ containing $n$ tokens, a model-generated story $s = s_1, ..., s_m$ containing $m$ tokens, and a perturbed story $s' = s'_1, ..., s'_{m'}$ containing $m'$ tokens, DELTASCORE calculates the likelihood difference under a language model:

$$\text{DELTASCORE}(s) = \log p(s|c) - \log p(s'|c) \quad (1)$$

where $p(s|c)$ represents the likelihood of $s$ conditioned on $c$ under a language model. In our experiments, we investigate several PLMs with varying architectures (§ 3.1) and perturbation techniques that are designed to target specific aspects (§ 3.2).

### 3.1   Two Different Likelihood Calculations

We now explain how we compute $p(s|c)$ with encoder-decoder PLMs (e.g., BART (Lewis et al., 2020) and T5 (Raffel et al., 2020)) and decoder PLMs (e.g., GPT-3 (Brown et al., 2020)). $p(s'|c)$ is computed in the same way and we omit it for brevity.

Denoting language model parameters as $\theta$, we compute DELTASCORE as follows for encoder-decoder PLMs:

$$\log p(s|c) = \frac{1}{m} \sum_{t=1}^{m} \log p(s_t|s_{<t}, c, \theta) \quad (2)$$

where $t$ denotes timestep in the sequence, and $s_{<t}$ denotes all tokens before the current timestep. Intuitively, the story condition $c$ is captured by the encoder, and the likelihood of the story $s$ is produced by the decoder.

In terms of decoder PLMs, we concatenate $c$ and $s$ to form a sequence $x$ ($x_1, ..., x_{n+m} = c_1, ..., c_n, s_1, ..., s_m$) to compute DELTASCORE:

$$\log p(s|c) = \frac{1}{m} \sum_{t=n+1}^{n+m} \log p(x_t|x_{<t}, \theta) \quad (3)$$

This formulation means we feed the full sequence including the story condition $c$ and story $s$ as input to the decoder-only PLM, although when computing the story likelihood, we only consider the conditional probabilities for the $s$ tokens.

### 3.2   Perturbations on Story Aspects

We follow Xie et al. (2023) to assess five fundamental aspects of story quality: fluency, coherence, relatedness, logicality, and interestingness. To this end, we survey perturbation methods from the literature (Ribeiro et al., 2020; Sai et al., 2021; Guan et al., 2021b; He et al., 2022) and attempt to align them to one of these five aspects. For some aspects, we also propose new perturbation methods. We now describe each aspect and its associated perturbation methods; A summary of these methods and examples is given in Table 1.

**Fluency**   assesses the readability of sentences in the story. Perturbations targeting fluency modify the text at the word or phrase level. We use two perturbation approaches from Ribeiro et al. (2020): 1) *Typo*, where we randomly transpose a character with an adjacent one in the text, and 2) *Subject-verb disagreement (SubjVerbDis)*, where we modify the verbs in a sentence so that they no longer agree with their subjects.

**Coherence**   assesses the level of connectivity between sentences in the story. Perturbations targeting coherence modify the text at the sentence level. We use two perturbation approaches from Sai et al. (2021): 1) *Jumble*, where we randomly shuffle words within the story, and 2) *Sentence Reorder (SentReorder)*, where we randomly shuffle the sentences within the story.

**Relatedness**   focuses on the extent to which the story is relevant to the given condition (e.g., story title). Perturbations targeting relatedness alter the story to reduce its association with its condition. We propose two new methods: 1) *Remove Relevant Words (RmRelWords)*, where we use ChatGPT[2] to identify words related to the given title and

---

[2]https://chat.openai.com

| Aspect | Perturbation | Original story | Perturbed story |
|--------|--------------|----------------|-----------------|
| Flu. | Typo | he went to see what the problem was | he went to see whta the problem was |
| | SubjVerbDis | he is the best student in the classroom . | he am the best student in the classroom . |
| Coh. | Jumble | We play badminton every evening . | badminton every We evening play . |
| | SentReorder | she did n't intend to buy anything . unfortunately she has poor impulse control ... | unfortunately she has poor impulse control . she did n't intend to buy anything ... |
| Rel. | RmRelWords | The supermarket has various kinds of goods | The has various kinds of goods |
| | StoryReplace | The supermarket has various kinds of goods | It is a nice day to hang out |
| Log. | Antonym | The boy got the gift he always wanted, he was so happy . | The boy got the gift he always wanted, he was so sad . |
| | Commonsense | they took me down to the lake . i threw my line out and caught several worms ... | they took me to the moon. i threw my line out and caught several stars ... |
| Int. | BlanderNarrative | i felt really angry, talked to my estranged father , and he gave me a gun! But I knew violence is not a solution here . | I felt upset and talked to my father about it . He advised me to handle the situation calmly , so I decided not to resort to violence . |

Table 1: Summary of perturbations that target a story quality aspect: Fluency (Flu.), Coherence (Coh.), Relatedness (Rel.), Logicality (Log.), and Interestingness (Int.). For "Relatedness" example stories, they are conditioned on the title "I always go to the local supermarket". Underlined perturbations are original methods we propose.

then remove them from the story, and 2) *Story Replacement (StoryReplace)*, where we substitute the original story with another story from a different story condition. To select a "comparable" story, we choose a story with where its likelihood is similar to the original story.[3]

**Logicality** focuses on the extent to which the story complies with commonsense. Perturbations targeting logicality introduce elements into the story that contradict commonsense. We adopt one approach from Guan et al. (2021b): *Antonym*, where we randomly replace the word with its antonym; and propose a new approach: *Commonsense*, where we use ChatGPT to modify some story elements to violate commonsense.

**Interestingness** measures the degree of predictability in the progression of events within a story, representing a highly subjective aspect. We propose one approach: *BlanderNarrative*, where we use ChatGPT to modify a story to make the narrative less interesting.

The ChatGPT[4] instructions for the aforementioned perturbations are detailed in Appendix A. For *Typo*, *Jumbo* and *Antonym*, we can control the degree of perturbation, and this parameter is tuned in § 5.1.

| Dataset | Condition | Story |
|---------|-----------|-------|
| ROC | [FEMALE] dad took me fishing . | we sat in a spot and waited for days ... |
| WP | tell me a story where the first line and last line ... | as i walked into the house , i was assailed by the smell of aging ... |

Table 2: Sampled examples of given story condition and its generated story for each dataset.

## 4 Experiments

### 4.1 Benchmarks

We use the generated stories and human ratings collected by Xie et al. (2023) on two story datasets: ROCStories (ROC; Mostafazadeh et al. (2016); 5-sentence simple stories) and WritingPrompts (WP; Fan et al. (2018); longer fictional stories written by users on Reddit).[5] The story condition ($c$) for ROC is the leading sentence; for WP, it is the short paragraph that describes the idea of the story, which is called "prompt". We present two example stories from the two datasets in Table 2.

Xie et al. (2023) experiment with 6 story generation models that cover large models with prompt-based learning (e.g., GPT-3), smaller fine-tuned models (e.g., BART) and other methods that in-

[3] We calculate the likelihood of the original story and a candidate story without considering their story conditions.

[4] We use OpenAI API with the model gpt-3.5-turbo.

[5] Xie et al. (2023) also collected human judgements for CNN-Dailymail, but we do not use them in our study for two reasons: 1) the stories depict real-world events rather than fictional narratives, and 2) most of the language models we test have been trained on this dataset, and so there are potential circularity issues.

corporate planning and commonsense (Xu et al., 2020; Guan et al., 2020, 2021a; Tan et al., 2021). They then conduct human evaluation on five aspects, judged using an ordinal scale from 1 (worst) to 5 (best). Two distinct groups of annotators were recruited, comprising in-house PhD students and crowdworkers. The results obtained from both groups were found to be similar, indicating the robustness and reliability of the annotation process.[6]

The judgment from the first group is used for preliminary exploration of optimal settings, such as assessing the effectiveness perturbation methods and language models (§ 5.1). The judgment of the second group is used for the final comparison of our approach with existing evaluation metrics (§ 5.2).

## 4.2 Language Models

We select a set of representative PLMs to compute DELTASCORE. For encoder-decoder PLMs, we use BART and FLAN-T5 (Chung et al., 2022). For decoder PLMs, we use BLOOM (Scao et al., 2022), LLaMA (Touvron et al., 2023), OPT (Zhang et al., 2022a), and GPT-3.5.[7] We use the largest possible variant whenever possible as we found larger models tend to work better in preliminary experiments. We present a summary of these models in Table 3.

## 4.3 Compared Evaluation Metrics

To comprehensively compare DELTASCORE with other existing evaluation metrics, we select representative evaluation metrics from each of the three categories mentioned in § 2.1.

For similarity metrics, we run experiments for **BLEU**, **BERTScore** and **MoverScore**. For discriminative metrics, we have **UNION**, **MANPLTS**, **StoryER**, **CTC** and **UNIEVAL**. Additionally, we include a zero-shot GPT-3.5 using simple instructions as prompt to gather judgements for the five specific aspects (Chiang and Lee, 2023). This approach is referred to as **GPT3.5Eval**; detailed instructions can be found in Appendix D.

Since UNION, MANPLTS and StoryER are all originally designed for story evaluation, we use their released models without fine-tuning for our experiments. For CTC, we use the reference-free

---

[6]For both groups, they gather assessments for 140 stories for ROC and 100 stories for WP, with each story being evaluated by three annotators. Annotations on human-written stories are excluded as they could introduce bias in favor of reference-based metrics. As a result, this leaves us with 120 stories for ROC and 80 stories for WP, respectively.

[7]We use text-davinci-003 in our experiments.

| Arch. | Model | Size | #Data | Objectives |
|---|---|---|---|---|
| En-De | BART | 406M | 160GB | Denoising |
| | FLAN-T5 | 11B | - | Denoising |
| De | BLOOM | 7B | 366BT | LM |
| | LLaMA | 65B | 1.4TT | LM |
| | OPT | 66B | 180BT | LM |
| | GPT-3.5 | 175B | 300BT | LM |

Table 3: Summary of PLMs, classified by their architecture (Arch.) as encoder-decoder (En-De) or decoder (De). "Size" indicates model parameters. #Data indicates the pre-trained data scale ("GB" = "gigabyte"; "BT" = "billion tokens"; and "TT" = "trillion tokens"). "LM" indicates causal language modeling objective.

alignment approach, which is also called "consistency" in the original paper. For UNIEVAL, the question answering models are trained on text summarization and dialogue generation tasks. We modify the questions to adapt UNIEVAL for evaluating different aspects of stories as the authors demonstrate the zero-shot transfer capability. Please refer to Appendix B for our questions. For generative metrics, we select **BARTScore** and **GPTScore**. We use the reference-free version of BARTScore (i.e., $c \rightarrow s$), and employ text-davinci-003 from OpenAI as the backbone of GPTScore with specific prompts for different story aspects. Prompts for GPTScore can be found in Appendix C.

We summarise all these metrics in Table 4, showing whether they: require additional training or ground truth reference; are originally introduced for story evaluation; and can measure fine-grained story aspects.

## 5 Results

We evaluate Kendall correlation at the story level, which involves comparing the predicted metric score versus the aggregated human rating for each story on a specific aspect. We use this as our primary metric due to the non-linear relationship between automatic and human metrics, as well as the ordinal scale employed in human judgments (Kendall, 1938). We explore different settings of our approach in § 5.1 and present a comparison of our best approach with existing evaluation metrics in § 5.2. Note that we use two different set of judgments, as explained in § 4.1, to avoid tuning and testing on the same test set.

| Objective | Metric | FT | B/F | ST | MS |
|---|---|---|---|---|---|
| Similarity | BLEU | ✗ | B | ✗ | ✗ |
| | BERTScore | ✗ | B | ✗ | ✗ |
| | MoverScore | ✗ | B | ✗ | ✗ |
| Discriminative | UNION | ✓ | F | ✓ | ✗ |
| | MANPLTS | ✓ | F | ✓ | ✗ |
| | StoryER | ✓ | F | ✓ | ✗ |
| | CTC | ✓ | F | ✗ | ✓ |
| | UNIEVAL | ✓ | F | ✗ | ✓ |
| Generative | BARTScore | ✗ | F | ✗ | ✓ |
| | GPTScore | ✗ | F | ✗ | ✓ |
| N/A | GPT3.5Eval | ✗ | F | ✗ | ✓ |

Table 4: Statistics of compared evaluation metrics. "FT" indicates whether the metric requires additional synthetic data to fine-tune on. "B/F" indicates whether the metric is reference-based (B) or reference-free (F). "ST" indicates whether the metric is originally designed for story evaluation. "MS" indicates whether the metric produces scores that consider multiple aspects.

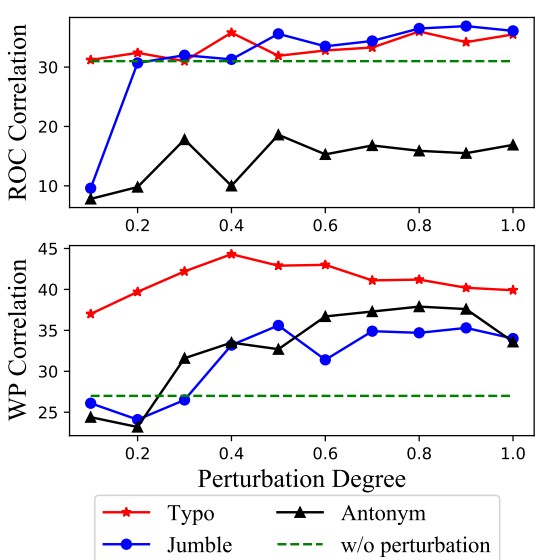

Figure 2: Impact of perturbation degree with LLaMA on in-house judgements for measuring coherence.

### 5.1 Preliminary Exploration

**Perturbation Methods** We commence by showcasing the comparative performance of various perturbation methods (§ 3.2) in relation to human judgments across the five aspects, as demonstrated in Table 5. For this analysis, we employ LLaMA as the PLM. The notation "w/o perturbation" denotes the calculation of story likelihood directly under LLaMA, without any perturbations applied. Our findings revealed intriguing results. Notably, we observed that perturbations specifically designed to target a particular aspect did not consistently exhibit a higher correlation with human judgments for that aspect. Furthermore, our analysis indicates that measuring interestingness is particularly challenging, as the correlation numbers associated with this aspect are generally lower compared to the other aspects. Finally, our last and perhaps most surprising observation is that a small set of perturbation methods, namely *Typo*, *Jumble*, and *Antonym*, exhibit strong performance in evaluating most aspects.

**Perturbation Degree** In the preceding phase, we carried out the most intense perturbation for *Jumble* and *Antonym*, in which the perturbation was applied to the entire sentence, and random selection of half of the characters for *Typo*. In light of their strong performance, we now investigate impact of perturbation degree using the these perturbation methods and present the results over ROC and WP in Figure 2. In the case of *Typo*, the degree pertains to the percentage of characters that we opt to swap. Concerning *Jumble*, we shuffle tokens within a certain text span while the span length is controlled by the degree. As for *Antonym*, we replace the token with its antonym under the specified probability (degree). As before, we use LLaMA as the PLM and focus on evaluating coherence here. Interestingly, *Typo* appears to be relatively stable and unaffected by the perturbation degree, where else *Jumble* and *Antonym* work better with more aggressive perturbation. Based on these results, we set the perturbation degree to 0.4, 0.9, and 0.8 for *Typo*, *Jumble*, and *Antonym* respectively for both ROC and WP.[8]

**Language Models** We next present DELTAS-CORE results using different PLMs in Table 6. We use the top 3 performing methods with the optimal degrees determined in our previous analysis. Encouragingly, across different PLMs and story aspects, we see that DELTASCORE outperforms vanilla likelihood ("w/o perturbation") in almost all instances, suggesting that measuring story quality using *likelihood difference* is generally a better approach than using its *likelihood* directly. Broadly speaking, *Jumble* is the most consistent perturbation method: in ROC it is the consistently the best performer, while in WP it is either the best or sec-

---

[8]The results in the previous subsection (Table 5) use these perturbation values.

| Target Aspect | Perturbation | ROC | | | | | WP | | | | |
|---|---|---|---|---|---|---|---|---|---|---|---|
| | | Flu. | Coh. | Rel. | Log. | Int. | Flu. | Coh. | Rel. | Log. | Int. |
| N/A | w/o perturbation | 17.3 | 31.0 | 22.8 | 35.2 | 20.2 | 26.8 | 27.0 | 32.1 | 34.6 | 17.8 |
| Fluency | Typo | 14.2 | 31.9 | 23.5 | 36.0 | 23.2 | 36.5 | 42.9 | 38.1 | 47.8 | 35.5 |
| | SubjVerbDis | 5.5 | 13.9 | 8.8 | 13.0 | 7.8 | 13.0 | 15.9 | 4.7 | 10.7 | 6.1 |
| Coherence | Jumble | 17.6 | 37.1 | 16.8 | 34.1 | 22.5 | 35.9 | 36.0 | 31.9 | 36.3 | 28.1 |
| | SentReorder | 4.4 | 17.7 | 4.8 | 20.3 | 8.9 | 18.6 | 27.3 | 20.5 | 15.1 | 25.4 |
| Relatedness | RmRelWords | 14.7 | 30.9 | 25.6 | 32.6 | 21.7 | 7.1 | 31.9 | 35.4 | 32.8 | 16.1 |
| | StoryReplace | 1.8 | 7.0 | 16.0 | 21.3 | 8.6 | 28.5 | 26.6 | 27.6 | 38.5 | 23.8 |
| Logicality | Antonym | 14.4 | 16.9 | 11.4 | 16.2 | 11.6 | 31.6 | 33.5 | 35.0 | 35.0 | 25.3 |
| | Commonsense | 20.1 | 27.1 | 20.7 | 34.9 | 21.4 | 8.5 | 16.0 | 12.3 | 17.4 | 8.8 |
| Interestingness | BlanderNarrative | 8.1 | 19.3 | 2.1 | 15.1 | 3.9 | 14.7 | 13.3 | 8.4 | 18.1 | 9.5 |

Table 5: Story-level Kendall correlation ($|\tau|$) between DELTASCORE with LLaMA and in-house judgements. We highlight instances where DELTASCORE outperforms vanilla likelihood ("w/o perturbation").

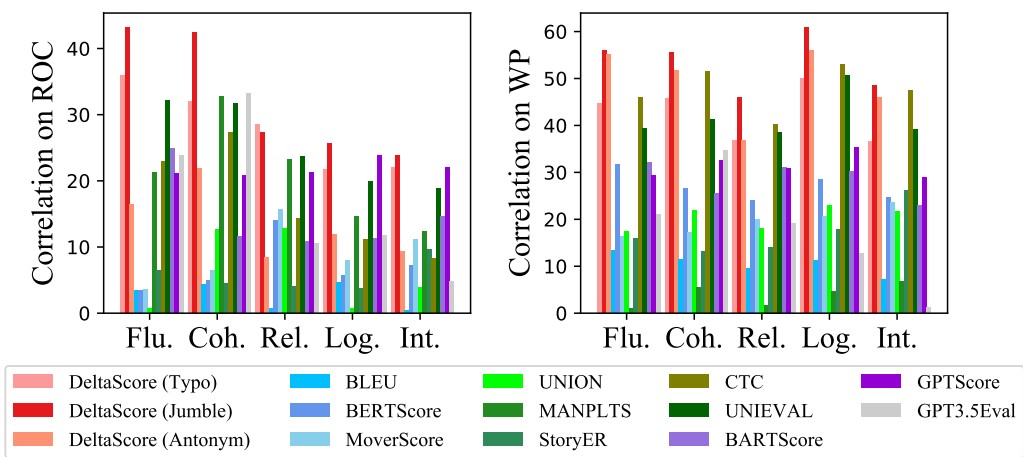

Figure 3: Absolute value of Story-level Kendall correlation ($|\tau|$) between different metrics and crowdworker ratings. Higher bar indicates better performance. Red bars indicate DELTASCORE. Blue bars indicate similarity based metrics. Green bars indicate discriminative metrics. Purple bars indicate generative metrics. Gray bar indicates GPT3.5Eval.

ond best performer, depending on the PLM. This observation aligns with the findings presented in Table 5, providing further confirmation that the *Jumble* perturbation method demonstrates effectiveness in measuring various story aspects. When examining the correlation magnitudes for different story aspects, it is evident that interestingness consistently exhibits lower values, reaffirming its inherent difficulty in measurement. There are, however, some curious exceptions: in ROC the correlation for fluency and relatedness is particularly low. We do not have a strong hypothesis of these observations, but will note that the language of ROC stories are somewhat formulaic and possibly different to the language of the pre-training data. For relatedness, the story condition in ROC is the first sentence, and it is a rather artificial condition to set

the "topic" for story generation.

An unsurprising observation is that larger models tend to exhibit stronger correlations, with GPT-3.5 and OPT performing the best among the PLMs. BLOOM and FLAN-T5 fall in the middle range, while BART shows the lowest correlation scores. Upon comparing GPT-3.5 and OPT, we observe a slight advantage for OPT despite its smaller model size and pre-training data. This finding suggests that beyond a certain scale, the benefits of further scaling may become less significant.

## 5.2 Comparison with Other Metrics

We next compare DELTASCORE with other evaluation metrics in Figure 3. Note that in this comparison, we utilize OPT as the chosen PLM, considering its superior performance, along with the same

| Metric | ROC | | | | | WP | | | | |
|---|---|---|---|---|---|---|---|---|---|---|
| | Flu. | Coh. | Rel. | Log. | Int. | Flu. | Coh. | Rel. | Log. | Int. |
| **DELTASCORE (with BART-large 406M)** | | | | | | | | | | |
| w/o perturbation | 4.5 | 14.6 | 11.6 | 14.5 | 2.3 | 15.0 | 14.5 | 22.4 | 24.5 | 10.6 |
| Jumble | 11.4 | 21.9 | 14.6 | 19.5 | 13.1 | 30.0 | 21.6 | 27.3 | 33.6 | 21.2 |
| Typo | 6.3 | 17.7 | 19.8 | 17.5 | 4.7 | 24.5 | 26.9 | 30.2 | 36.6 | 24.3 |
| Antonym | 5.7 | 8.3 | 7.3 | 5.4 | 6.9 | 28.0 | 26.4 | 37.6 | 31.8 | 25.7 |
| **DELTASCORE (with FLAN-T5 XXL 11B)** | | | | | | | | | | |
| w/o perturbation | 16.2 | 19.4 | 14.7 | 23.1 | 8.9 | 19.9 | 14.1 | 23.4 | 20.8 | 4.8 |
| Jumble | 24.5 | 36.1 | 20.2 | 27.9 | 20.6 | 31.7 | 28.3 | 22.1 | 34.4 | 21.8 |
| Typo | 11.3 | 21.8 | 11.2 | 28.0 | 14.7 | 29.0 | 25.7 | 33.1 | 31.8 | 18.0 |
| Antonym | 10.7 | 12.9 | 8.9 | 10.1 | 7.9 | 33.1 | 34.0 | 30.6 | 34.2 | 29.2 |
| **DELTASCORE (with BLOOM 7B)** | | | | | | | | | | |
| w/o perturbation | 12.4 | 24.1 | 13.3 | 25.5 | 10.2 | 24.8 | 21.3 | 27.9 | 29.2 | 12.6 |
| Jumble | 25.0 | 35.5 | 14.2 | 33.9 | 25.4 | 34.2 | 34.8 | 30.7 | 33.6 | 23.1 |
| Typo | 14.3 | 31.6 | 12.2 | 30.3 | 14.5 | 38.4 | 40.9 | 38.7 | 44.7 | 29.5 |
| Antonym | 10.2 | 12.0 | 9.4 | 10.3 | 12.2 | 27.4 | 31.5 | 31.3 | 32.9 | 30.6 |
| **DELTASCORE (with LLaMA 65B)** | | | | | | | | | | |
| w/o perturbation | 17.3 | 31.0 | 22.8 | 35.2 | 20.2 | 26.8 | 27.0 | 32.1 | 34.6 | 17.8 |
| Jumble | 19.7 | 36.9 | 15.8 | 35.9 | 23.0 | 36.8 | 35.3 | 32.9 | 35.5 | 27.7 |
| Typo | 14.2 | 31.9 | **23.5** | 36.0 | 23.2 | 34.3 | 42.2 | 35.2 | 44.0 | 31.8 |
| Antonym | 13.3 | 15.9 | 10.6 | 14.2 | 11.0 | 31.3 | 37.9 | 35.0 | 36.2 | 31.6 |
| **DELTASCORE (with OPT 66B)** | | | | | | | | | | |
| w/o perturbation | 17.4 | 30.6 | 20.2 | 32.6 | 15.8 | 27.5 | 24.5 | 32.2 | 31.0 | 15.7 |
| Jumble | **27.4** | **44.2** | 21.6 | **41.4** | **30.0** | 37.9 | 38.7 | 35.4 | 39.5 | 32.0 |
| Typo | 17.9 | 39.6 | 21.3 | 38.7 | 22.8 | 39.0 | 41.9 | **38.0** | **45.6** | 30.4 |
| Antonym | 15.8 | 19.0 | 13.8 | 14.9 | 12.9 | 34.0 | 40.3 | 36.0 | 39.4 | 36.7 |
| **DELTASCORE (with GPT-3.5 175B)** | | | | | | | | | | |
| w/o perturbation | 21.3 | 31.2 | 18.7 | 28.6 | 18.2 | 36.2 | 32.9 | 36.0 | 37.1 | 23.3 |
| Jumble | 18.7 | 34.0 | 23.3 | 38.5 | 29.7 | 35.0 | 36.6 | 36.1 | 37.7 | 33.5 |
| Typo | 17.0 | 35.1 | 11.7 | 32.2 | 26.3 | **41.7** | **46.7** | 37.5 | 45.0 | **37.3** |
| Antonym | 19.1 | 21.5 | 12.9 | 20.5 | 13.9 | 36.1 | 41.2 | 41.3 | 39.6 | 38.2 |

Table 6: Absolute value of Story-level Kendall correlation ($|\tau|$) between different metrics and in-house judgements. We **bold** the best scores for each aspect and highlight instances where DELTASCORE improves over vanilla likelihood ("w/o perturbation").

top-performing perturbation methods. The results of our evaluation are very promising: DELTAS-CORE consistently outperforms all competitor metrics across all story aspects. *Jumble* stands out as the most effective perturbation method among the three. The similarity metrics generally has the lowest performance, highlighting the inadequacy of reference-based metrics for story evaluation, which aligns with previous research findings (Guan and Huang, 2020; Xie et al., 2023). Among the discriminative metrics, CTC and UNIEVAL show relatively strong competitiveness, although they still fall behind DELTASCORE. The performance of generative scores is inconsistent. GPTScore shows strong performance in evaluating logicality and interestingness, especially in ROC, where it performs similarly to DELTASCORE. However, its effectiveness is limited in other scenarios. More detailed scores can be found in Table 8.

## 6 Discussion and Conclusion

Initially, our aim was to investigate various types of perturbations for assessing fine-grained aspects of storytelling. But seeing that performance of each metric does not vary much across different aspects in Figure 3, it suggests these aspects may be somewhat inter-correlated. Also, our findings revealed that one of the simplest perturbation methods, namely *Jumble*, is exceptionally effective in measuring most aspects. One hypothesis could be that *Jumble* is functioning as a *normalisation factor* to modulate word frequency and sentence length effects when estimating sequence quality. This finding aligns with prior study that used sequence probabilities for measuring sentence acceptability (Lau et al., 2020). They found that it is important to normalise the probabilities and introduced various normalization techniques to mitigate the impact of word frequency and sentence length. *Jumble* can

be interpreted as an alternative normalisation technique. Given this insight, it may also mean that DELTASCORE has broader application beyond the evaluation of story quality. For instance, it could be used to score sentences in machine translation and summarization.

In conclusion, we introduce DELTASCORE, a novel approach to assess fine-grained story aspects by comparing the likelihood difference between the original story and a perturbed version using a pre-trained language model. Surprisingly, we discovered that a small set of perturbation methods excel in measuring the majority of story aspects. Furthermore, our findings demonstrate that DELTASCORE shows stronger correlations with human judgments compared to a range of existing metrics across two different story domains.

## Limitations

Our study only investigates a limited range of perturbations, and we acknowledge that there may be other forms of perturbations that could work better. The field of evaluation metrics is rapidly evolving, with numerous contemporary evaluation metrics introduced recently, such as G-Eval (Liu et al., 2023) and ChatEval (Chan et al., 2023), which were not incorporated into the comparative evaluation metrics within this study.

## Acknowledgements

We extend our thanks to the reviewers for their valuable feedback, which has greatly contributed to the improvement of this work. Zhuohan Xie is supported by Melbourne Research Scholarship, and would like to expresses his sincere appreciation to the program.

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

## A Perturbation Prompts

We use following prompts for perturbations when we apply API with GPT-3.5-turbo.

- RelevantWords: Find all words in the given story that is relevant to the given title. Please only print words in the given story, and separate them by ','. "title": {title}, "story": {story}

- Commonsense: Revise the following story such that certain elements does not make sense. The revision should be minimal, e.g., by changing a few words. "story": {story}

- BlanderNarrative: Revise the following story to make it less interesting (e.g., expected ending, no plot twist). The revision should be minimal. "story": {story}

We present several examples of perturbed stories by GPT-3.5-turbo in Table 7. We observe that *BlanderNarrative* does not significantly alter the original story. This observation is in line with previous findings that *BlanderNarrative* does not effectively impact interestingness in Table 5. We speculate that this outcome may be attributed to the inherent simplicity of most stories, which limits the extent to which GPT-3.5-turbo can modify them to reduce their level of interest.

## B UNIEVAL Questions

We ask the following questions for each aspect. Note that we try to use the narrative/vocabulary as close to the original questions Zhong et al. (2022) use in their efforts as possible.

- Fluency: Is this a fluent utterance?

- Coherence: Is this a coherent utterance?

- Relatedness: Is this claim consistent with the document?

- Logicality: Is this utterance consistent with the commonsense?

- Interestingness: Is this an interesting utterance?

## C GPTScore Prompts

We use the following prompts for each aspect. Note that we try to use the narrative/vocabulary as close to the original prompts (Fu et al., 2023) use in their efforts as possible.

| Perturbation | Title | Original Story | Perturbed Story |
|---|---|---|---|
| RmRelWords | the man ordered extra cheese . | the man loved cheese and always ordered extra . his favourite was pizza with extra cheese . one day , he went to his favourite pizza place and ordered his usual . when the pizza arrived , he saw that there was no extra cheese . the man was so angry that he threw the pizza across the room . | man loved and always . his favourite was . one day , he went to his favourite place and ordered his usual . when the pizza arrived , he saw that there was . the man was so angry that he the pizza across the room . |
| RmRelWords | the boy cried . | the boy cried because he was sad and scared . his parents were getting a divorce and he did n't know what was going to happen to him . he was going to have to leave his home and his friends and move to a new house . his mother said that he could choose where he wanted to live , but his father said that he had to live with him . the boy did n't know what to do . | because was and . were getting a and he was going happen . he was going to have to his and his friends and to a . his mother said that he could he to , but his father said that he had to him . the boy did n't know what to . |
| Commonsense | [FEMALE] dad took me fishing . | they took me down to the lake . i threw my line out and caught several worms . i turned in one worm and caught a catfish . i told my dad and he took me home and i raised it for dinner . | they took me down to the sky . i threw my book out and caught several words . i turned in one word and caught a spaceship . i told my mom and she took me home , and i raised it for dinner . |
| Commonsense | [FEMALE] was at the mall . | she was walking to the food court when she saw a man who looked lost . she went up to him and asked if he needed help . the man told her he was looking for his wife and daughter . [FEMALE] took him to the food court and pointed them out to him . | she was flying to the food court when she saw a bird who looked lost . she went up to it and asked if it needed help . the bird told her it was looking for its wife and daughter . [FEMALE] took it to the food court and pointed them out to it . |
| BlanderNarrative | [FEMALE] went out to eat with her friends . | she ordered a burger and fries , but when the food arrived , she was really disappointed . the burger was tiny and the fries were cold and soggy . | she ordered a burger and fries . when the food arrived , it was just as she expected . the burger was small and the fries were lukewarm . |
| BlanderNarrative | [MALE] is out with his friends at the bar . | he decides to buy a beer . [MALE] drinks a beer and eats a few more . [MALE] feels very sick . [MALE] is embarrassed that he drank so much . | he decides to buy a beer . [MALE] drinks a beer and eats a few more . [MALE] feels sick . [MALE] regrets drinking too much . |

Table 7: We show some examples of perturbed stories where we use GPT-3.5-turbo for perturbation.

- Fluency: Generate a fluent story for the given title: {title}, and story: {story}

- Coherence: Generate a coherent story for the given title: {title}, and story: {story}

- Relatedness: Generate a story related to the given title: {title}, and story: {story}

- Logicality: Generate a story that adhere to commonsense for the given title: {title}, and story: {story}

- Interestingness: Generate an interesting story for the given title: {title}, and story: {story}

## D  GPT3.5Eval Instructions

We use the following instructions for each aspect, following Chiang and Lee (2023).

**Fluency**   The goal of this task is to rate story fragment.
Note: Please take the time to fully read and understand the story fragment.
Story fragment:
{{Story}}
How fluent is the text of the story fragment? (on a scale of 1-5, with 1 being the lowest)

**Coherence**   The goal of this task is to rate story fragment.
Note: Please take the time to fully read and understand the story fragment.
Story fragment:
{{Story}}
How coherent is the text of the story fragment? (on a scale of 1-5, with 1 being the lowest)

**Relatedness**   The goal of this task is to rate story fragment.

Note: Please take the time to fully read and understand the story fragment.

Story title:

{{Title}}

Story fragment:

{{Story}}

How related is the text of the story fragment to the title? (on a scale of 1-5, with 1 being the lowest)

**Logicality**   The goal of this task is to rate story fragment.

Note: Please take the time to fully read and understand the story fragment.

Story fragment:

{{Story}}

How logically correct is the text of the story fragment? (on a scale of 1-5, with 1 being the lowest)

**Interestingness**   The goal of this task is to rate story fragment.

Note: Please take the time to fully read and understand the story fragment.

Story fragment:

{{Story}}

How interesting is the text of the story fragment? (on a scale of 1-5, with 1 being the lowest)

| Metric | ROC | | | | | WP | | | | |
|---|---|---|---|---|---|---|---|---|---|---|
| | **Flu.** | **Coh.** | **Rel.** | **Log.** | **Int.** | **Flu.** | **Coh.** | **Rel.** | **Log.** | **Int.** |
| **Similarity Based Metrics** | | | | | | | | | | |
| BLEU | 3.4 | 4.4 | 0.8 | 4.6 | 0.4 | 13.4 | 11.4 | 9.6 | 11.2 | 7.2 |
| BERTScore | 3.5 | 5.0 | 14.0 | 5.7 | 7.3 | 31.8 | 26.7 | 24.0 | 28.5 | 24.6 |
| MoverScore | 3.6 | 6.5 | 15.7 | 8.0 | 11.2 | 16.4 | 17.2 | 20.1 | 20.7 | 23.7 |
| **Discriminative Metrics** | | | | | | | | | | |
| UNION | 0.7 | 12.7 | 12.8 | 0.8 | 3.9 | 17.4 | 21.9 | 18.1 | 22.9 | 21.8 |
| MANPLTS | 21.3 | 32.8 | 23.2 | 14.7 | 12.4 | 1.1 | 5.5 | 1.7 | 4.6 | 6.7 |
| StoryER | 6.5 | 4.5 | 4.0 | 3.7 | 9.7 | 15.9 | 13.1 | 14.1 | 17.9 | 26.1 |
| CTC | 22.9 | 27.3 | 14.3 | 11.1 | 8.3 | 45.9 | 51.6 | 40.3 | 53.1 | 47.5 |
| UNIEVAL | 32.2 | 31.7 | 23.7 | 20.0 | 18.8 | 39.3 | 41.3 | 38.6 | 50.7 | 39.1 |
| **Generative Metrics** | | | | | | | | | | |
| BARTScore | 24.9 | 11.6 | 10.9 | 11.3 | 14.6 | 32.2 | 25.6 | 31.0 | 30.2 | 23.0 |
| GPTScore | 21.2 | 20.9 | 21.3 | 23.9 | 22.0 | 29.3 | 32.6 | 30.8 | 35.3 | 28.9 |
| GPT3.5Eval | 23.9 | 33.3 | 10.5 | 11.8 | 4.8 | 21.1 | 34.6 | 19.2 | 12.8 | 1.2 |
| **DELTASCORE** | | | | | | | | | | |
| Typo | 35.9 | 32.0 | **28.6** | 21.8 | 22.1 | 44.8 | 45.8 | 36.8 | 50.1 | 36.6 |
| Jumble | **43.2** | **42.4** | 27.4 | **25.6** | **23.9** | **56.1** | **55.5** | **45.9** | **60.9** | **48.5** |
| Antonym | 16.5 | 21.9 | 8.4 | 11.9 | 9.3 | 55.2 | 51.8 | 36.9 | 56.0 | 45.9 |

Table 8: Absolute value of Story-level Kendall correlation ($|\tau|$) between different metrics and crowdworker ratings. We **bold** the best scores in each aspect.