# OpenReview forum: "DeltaScore: Fine-Grained Story Evaluation with Perturbations"
_EMNLP/2023/Conference — EMNLP 2023 Findings_

### Official Review · Reviewer_cnBq · 2023-08-04

**Soundness:** 4

**Excitement:**

4: Strong: This paper deepens the understanding of some phenomenon or lowers the barriers to an existing research direction.

**Paper Topic And Main Contributions:**

This paper presented a new evaluation method for model-generated narratives by comparing the likelihood of the next tokens from the original and the perturbed stories. The authors showed that their methods correlate better with human judgments than other conventional NLG evaluation metrics across five domains. The evaluation results were able to cover fine-grain and relatively simple to use.

**Questions For The Authors:**

A. Did you find any bias in the perturbation results from ChatGPT? For example, how ChatGPT selected related words or stories for the Relatedness domain or revised commonsense output?

**Reasons To Accept:**

- Thorough analysis: The paper tested their method across five different domains, using two perturbation methods each, including four newly introduced methods from the related studies. The results extensively compared their method with multiple existing methods for NLG evaluations. The analysis also covered the results from various encoder-decoder and decoder-only PLMs.
- The methods presented in the paper were simple but effective.
- Good presentation: Figures and Tables were effective. Reading the manuscript was enjoyable.


**Reasons To Reject:**

I'd like to see further clarification on the quality of the perturbed/augmented data.
- The quality of perturbation outputs was not validated. For example, the Blander Narrative method was shown as the least effective method, but we cannot be sure if that was due to the ChatGPT's output quality for "making the narrative less interesting."
- Perturbation methods in Fluency would need further clarifications. For example, how did you select a character/word to alter?

**Reproducibility:**

3: Could reproduce the results with some difficulty. The settings of parameters are underspecified or subjectively determined; the training/evaluation data are not widely available.

**Reviewer Confidence:**

3: Pretty sure, but there's a chance I missed something. Although I have a good feel for this area in general, I did not carefully check the paper's details, e.g., the math, experimental design, or novelty.

---

> ### Author Rebuttal · Authors · 2023-08-28
>
> Thank you for providing your review, we have following responses regarding the issues/questions you point out.
>
> **Responses To Issues:**
>
> _1. The quality of perturbation outputs was not validated. For example, the Blander Narrative method was shown as the least effective method, but we cannot be sure if that was due to the ChatGPT's output quality for "making the narrative less interesting."_
>
> We did some analyses in preliminary experiments to engineer/tune the instructions (to make sure the approach is optimal), but we agree that this can be validated more rigorously. We will perform error analyses on the generated output and include them in the revision.
>
> _2. Perturbation methods in Fluency would need further clarifications. For example, how did you select a character/word to alter?_
>
> We use perturbation approaches provided by checklist (Ribeiro et al., 2020) and openmeva (Guan et al., 2021).
> For typo, we choose certain number of characters to swap, number = story length * perturbation degree; for jumble, we choose a span of random length to shuffle tokens, length = story length * perturbation degree; for antonym, we change the token to its antonym using wordnet under the certain probability (perturbation degree).
> We will include these details in the revision.
>
> **Responses To Questions:**
>
> *1. Did you find any bias in the perturbation results from ChatGPT? For example, how ChatGPT selected related words or stories for the Relatedness domain or revised commonsense output?*
>
> For commonsense, it usually changes some entities to the unrelated ones to break the logic, such as "she put the batter in the oven and waited for it to bake ." -> "she put the hat in the tennis court and waited for it to bake .".
> We did not notice any obvious patterns in terms of word selection, but agree it is possible that there could be some kinds of bias (e.g., cultural or domain or others) with a deeper analysis. We will include random example generations in the main paper (to help reader have a grasp on the overall quality) and release the full output for the community to enable future research in this direction.

---

### Official Review · Reviewer_pCUx · 2023-08-05

**Soundness:** 4

**Excitement:**

4: Strong: This paper deepens the understanding of some phenomenon or lowers the barriers to an existing research direction.

**Paper Topic And Main Contributions:**

This works proposes DeltaScore, an unsupervised metric for story generation. It is based on measuring difference in likelihood between a generated story and a specifically perturbed version of the same story under a generic pre-trained language model (PLM). The work proposes multiple perturbation methods targeting 5 aspects of evaluation: fluency, coherence, relatedness, logicality, and interestingness.
They evaluate DeltaScore on two story generation datasets and compute correlation to human judgments based on annotations from Xie et a. (2023). While most of the proposed perturbations do not consistently improve correlations for the specific aspect they target, the authors identify 3 perturbation methods that improve all aspects. The DeltaScore methods was tested with multiple PLM backbones and compared to   various existing evaluation metrics.

**Reasons To Accept:**

Overall I found the work to be very well written and presented. The arguments flow well and choices are well argued.

The method itself is well described and seems relatively simple to use and robust with respect to the choice of backbone PLM (see Table 6).

I like that the evaluation procedure uses separate sets of human ratings for the preliminary study of parameter choices and the full comparison across PLM and with other metrics.

The initial goal seemed to be to have specific targeted perturbations for each aspect, which resulted in a mixed outcome (Table 5). I appreciate the attempt to explain the surprising performance of "jumble" across all aspects in Section 6.

**Reasons To Reject:**

While the method works across all aspects for a subset of perturbations, the initial goal of finding a targeted method for individual aspects remains open. The method seems to work empirically but it is unclear why.

On lines L375-L386, talking about the perturbation degrees, it is not entirely clear what those numbers mean. A practitioner can likely guess what they are supposed to mean but it would be better to be explicit.

Similarly, it is not clear what perturbation parameters were used in Table 5.

**Reproducibility:**

4: Could mostly reproduce the results, but there may be some variation because of sample variance or minor variations in their interpretation of the protocol or method.

**Reviewer Confidence:**

3: Pretty sure, but there's a chance I missed something. Although I have a good feel for this area in general, I did not carefully check the paper's details, e.g., the math, experimental design, or novelty.

**Typos Grammar Style And Presentation Improvements:**

L217: "substitute" -> I'd suggest saying "transpose" or "exchange" as "substitute" could be interpreted as a doubling of a letter

I think it would be helpful to include a table of dataset statistics such as number of annotated stories and number of raters for the data of Xie et al. (2023).

I think Fig. 3 would be a lot better as a table! As is, it is very difficult to glean more than a very high-level idea of the comparison.

---

> ### Author Rebuttal · Authors · 2023-08-28
>
> Thank you for providing your review, we have following responses regarding the issues/questions you point out.
>
> **Responses To Issues:**
>
> *1. While the method works across all aspects for a subset of perturbations, the initial goal of finding a targeted method for individual aspects remains open. The method seems to work empirically but it is unclear why.*
>
> We agree that it is a curious finding. In section 6 we discussed and hypothesised that jumble might be functioning as a normalisation factor to regulate word frequency and sentence length for estimating quality, thus it provides great performance with DeltaScore.
> It is true that finding a targeted method for individual aspects remain open as it is trikcy to find approaches to target on one aspect while keeping the others the same, and this paves way for future theoretical work/analyses.
>
> *2. On lines L375-L386, talking about the perturbation degrees, it is not entirely clear what those numbers mean. A practitioner can likely guess what they are supposed to mean but it would be better to be explicit.*
>
> We use perturbation approaches provided by the checklist (Ribeiro et al., 2020) and openmeva papers (Guan et al., 2021).
> For typo, we choose certain number of characters to swap, number = story length * perturbation degree; for jumble, we choose a span of a random length to shuffle tokens, length = story length * perturbation degree; for antonym, we change the token to its antonym using wordnet under the certain probability (perturbation degree).
>
> *3. Similarly, it is not clear what perturbation parameters were used in Table 5.*
>
> For Table 5, we were selecting methods before selecting the best performing degrees, the degree for typo, jumble and antonym are 0.5, 1.0, and 1.0, respectively.
> We will include these details into the revision.
>
> **Responses To Presentation Improvements:**
>
> *1. L217: "substitute" -> I'd suggest saying "transpose" or "exchange" as "substitute" could be interpreted as a doubling of a letter*
>
> We will change the word in the revision, thank you for the suggestion.
>
> *2. I think it would be helpful to include a table of dataset statistics such as number of annotated stories and number of raters for the data of Xie et al. (2023).*
>
> We agree and will include this in the revision.
>
> *3. I think Fig. 3 would be a lot better as a table! As is, it is very difficult to glean more than a very high-level idea of the comparison.*
>
> Thanks for the suggestion. We will add the table in the revision.

---

### Official Review · Reviewer_5YuF · 2023-08-07

**Typos Grammar Style And Presentation Improvements:** line 258 instructions typo
**Soundness:** 3

**Excitement:**

3: Ambivalent: It has merits (e.g., it reports state-of-the-art results, the idea is nice), but there are key weaknesses (e.g., it describes incremental work), and it can significantly benefit from another round of revision. However, I won't object to accepting it if my co-reviewers champion it.

**Paper Topic And Main Contributions:**

- introduced DELTASCORE, a  method that employs perturbation techniques for the evaluation of nuanced story aspects.
- the central proposition is that the extent to which a story excels in a specific aspect (e.g., fluency) correlates with the magnitude of its susceptibility to particular perturbations (e.g., the introduction of typos).
- measuring the quality of an aspect by calculating the likelihood difference between pre- and post-perturbation states using pre-trained LMs.

**Questions For The Authors:**

1. see the above reasons to reject
2. it seems that the likelihood regarding the frequency of typos are non-linear. any empirical validations?
3. any other works also used perturbation for evaluation purposes in NLP field in general?

**Reasons To Accept:**

- comprehensive experiments and compared with a set of baselines


**Reasons To Reject:**

- lacks of intuition of the connection between why perturbation-based evaluation could be  useful and the model working mechanism or human evaluation. Although the effectiveness is validated empirically. It would be helpful to see more discussion related to why this perturbation-based method is useful.
- it would be interesting to include a baseline that is zero shot prompting gpt3.5 asking the model about the 5 aspects and compare with human annotations.


**Reproducibility:**

3: Could reproduce the results with some difficulty. The settings of parameters are underspecified or subjectively determined; the training/evaluation data are not widely available.

**Reviewer Confidence:**

2: Willing to defend my evaluation, but it is fairly likely that I missed some details, didn't understand some central points, or can't be sure about the novelty of the work.

---

> ### Author Rebuttal · Authors · 2023-08-28
>
> Thank you for providing your review, we have following responses regarding the issues/questions you point out.
>
> **Responses To Issues:**
>
> *1. lacks of intuition of the connection between why perturbation-based evaluation could be useful and the model working mechanism or human evaluation. Although the effectiveness is validated empirically. It would be helpful to see more discussion related to why this perturbation-based method is useful.*
>
> Our idea is mostly based on the intuition that higher quality text will be affected more by perturbations than lower quality text (shown in Figure 1) and our empirical results validate this. This perturbation-based mechanism also allows a straightforward way to inject inductive bias into evaluation metrics, namely that we expect some perturbations to have a more marked effect on a high versus low-quality texts.
>
> *2. it would be interesting to include a baseline that is zero shot prompting gpt3.5 asking the model about the 5 aspects and compare with human annotations.*
>
> Thank you for this suggestion. We have done some experiments for zero-shot prompting with simple instructions to ask GPT-3.5 to evaluate the stories on five aspects directly, on crowdsourced data which is described in Table 6. The Kendall correlation results are as follows.
> For ROC, we have correlations on fluency: 23.90, on coherence: 33.27, on relatedness: 10.49, on logicality: 11.82, on interestingess: 4.78
> For WP, we have correlations on fluency: 21.10, on coherence: 34.55, on relatedness: 19.20, on logicality: 12.81, on interestingess: 1.22
> This zero-shot prompting approach shows better performance in terms of fluency and coherence than recently proposed BERTScore and BARTScore, but it is still behind DeltaScore in all aspects.
> We will include this new set of results as another baseline in the revision.
>
> **Responses To Questions:**
>
> *1. see the above reasons to reject*
>
> See above responses.
>
> *2. it seems that the likelihood regarding the frequency of typos are non-linear. any empirical validations?*
>
> Assuming that you meant the relation between the correlation and the perturbation degree in Figure 2, then yes the relation appears to be non-linear. Intuitively, it may be because of the subword tokenisation. Introducing too many typos will disrupt subword tokenisation too much and result in unreliable probability estimates, hence the sweet spot between 0.4 and 0.6.
>
> *3. any other works also used perturbation for evaluation purposes in NLP field in general?*
>
> To the best of our knowledge, we were the first to use perturbations directly for evaluating generated texts by the time of writing this paper; however, there are other works such as Ribeiro et al., 2020 and Sai et al., 2021 use perturbations to assess the robustness of evaluation metrics. We discussed this in Sec 2.2 (line 144-149).
> Perturbations are also used in training LLMs, that is, masking, dropping and substituting tokens are commonly used, e.g., in T5 / BART style encoder-decoder approaches where the LLM is trained to **denoise** the input.
>
> **Responses To Presentation Improvements:**
>
> *1. line 258 instructions typo*
>
> Thank you for pointing this out, we will revise it.

---

### Meta-Review · Area_Chair_kr6Y · 2023-09-16

**Recommendation:** 4

**Metareview:**

The paper introduces a new evaluation method, DeltaScore, for evaluating nuanced aspects of story generation models. The authors propose using perturbation techniques to assess the quality of a story's specific aspect (e.g., fluency), suggesting that the susceptibility of a story to certain perturbations (e.g., the introduction of typos) correlates with the story's excellence in that aspect. The quality of an aspect is measured by calculating the likelihood difference between pre- and post-perturbation states using pre-trained language models (LMs).

The reviewers generally appreciated the comprehensive experiments and comparisons with baseline methodologies. They also acknowledged that the paper was well-written and presented, with effective use of figures and tables. The simplicity and effectiveness of the proposed method were also noted as strengths.

However, some concerns were raised about the lack of intuition regarding the connection between perturbation-based evaluation and the model working mechanism or human evaluation. Reviewers suggested that a more thorough discussion about why this method is useful would be beneficial. The authors were also asked about the possibility of bias in the perturbation results from ChatGPT, and how it selected related words or stories for the Relatedness domain or revised commonsense output. One of the reviewers also noted that the quality of perturbation outputs was not validated.

In response to these concerns, the authors provided thorough rebuttals. They discussed their intuition behind the perturbation-based method and presented additional experimental results to support their claims. They also clarified the mechanisms of their perturbation methods and promised to include more details in their revisions. The authors acknowledged the need for rigorous validation of perturbation outputs and committed to performing error analyses in the future.

Overall, the reviewers found the study to be generally strong and sound, providing sufficient support for its claims. They were excited about the paper and believed it could deepen the understanding of certain phenomena or lower the barriers to an existing research direction. However, they also acknowledged that there were areas for improvement and additional work needed, particularly in terms of validating the quality of perturbation outputs and clarifying the perturbation methods used.

---

### Decision · Program_Chairs · 2023-10-07

**Decision:**

Accept-Findings

**Comment:**

The paper introduces a new evaluation method, DeltaScore, for evaluating nuanced aspects of story generation models. The authors propose using perturbation techniques to assess the quality of a story's specific aspect (e.g., fluency), suggesting that the susceptibility of a story to certain perturbations (e.g., the introduction of typos) correlates with the story's excellence in that aspect. The quality of an aspect is measured by calculating the likelihood difference between pre- and post-perturbation states using pre-trained language models (LMs).

The reviewers generally appreciated the comprehensive experiments and comparisons with baseline methodologies. They also acknowledged that the paper was well-written and presented, with effective use of figures and tables. The simplicity and effectiveness of the proposed method were also noted as strengths.

However, some concerns were raised about the lack of intuition regarding the connection between perturbation-based evaluation and the model working mechanism or human evaluation. Reviewers suggested that a more thorough discussion about why this method is useful would be beneficial. The authors were also asked about the possibility of bias in the perturbation results from ChatGPT, and how it selected related words or stories for the Relatedness domain or revised commonsense output. One of the reviewers also noted that the quality of perturbation outputs was not validated.

In response to these concerns, the authors provided thorough rebuttals. They discussed their intuition behind the perturbation-based method and presented additional experimental results to support their claims. They also clarified the mechanisms of their perturbation methods and promised to include more details in their revisions. The authors acknowledged the need for rigorous validation of perturbation outputs and committed to performing error analyses in the future.

Overall, the reviewers found the study to be generally strong and sound, providing sufficient support for its claims. They were excited about the paper and believed it could deepen the understanding of certain phenomena or lower the barriers to an existing research direction. However, they also acknowledged that there were areas for improvement and additional work needed, particularly in terms of validating the quality of perturbation outputs and clarifying the perturbation methods used.